# A Hypoxia-Inducible Factor Stabilizer Improves Hematopoiesis and Iron Metabolism Early after Administration to Treat Anemia in Hemodialysis Patients

**DOI:** 10.3390/ijms21197153

**Published:** 2020-09-28

**Authors:** Chie Ogawa, Ken Tsuchiya, Naohisa Tomosugi, Kunimi Maeda

**Affiliations:** 1Maeda Institute of Renal Research, 6F-1-403 Kosugi-cho, Nakahara-ku, Kawasaki 211-0063, Japan; kuni@maeda-irr.com; 2Biomarker Society, INC, 6F-1-403 Kosugi-cho, Nakahara-ku, Kawasaki 211-0063, Japan; tsuchiya@twmu.ac.jp (K.T.); tomosugi@kanazawa-med.ac.jp (N.T.); 3Department of Blood Purification, Tokyo Women’s Medical University, 8-1 Kawada-cho, Shinjuku-ku, Tokyo 162-8666, Japan; 4Division of Systems Bioscience for Drug Discovery Project Research Center, Medical Research Institute, Kanazawa Medical University, 1-1 Daigaku, Uchinada-machi, Kahoku-gun, Ishikawa 920-0293, Japan

**Keywords:** hypoxia-inducible factor stabilizer, erythropoiesis, iron metabolism, hemodialysis, renal anemia

## Abstract

Roxadustat (Rox), a hypoxia-inducible factor (HIF) stabilizer, is now available for the treatment of anemia in hemodialysis (HD) patients. To investigate hematopoietic effect and iron metabolism, this study involved 30 HD patients who were initially treated with darbepoetin (DA), a conventional erythropoietin-stimulating agent, and then switched to Rox. We measured erythrocyte, reticulocyte indices, and iron-related factors at every HD during the first two weeks after the treatment switch (Days 0–14) and again on Days 21 and 28. We measured erythropoietin (EPO) concentration every week and examined their changes from Day-0 values. The same variables were measured in 15 HD patients who continued DA at every HD for one week. Iron-related factors were also measured on Days 14 and 28. In the Rox group, hepcidin significantly decreased from Day 2. The reticulocyte hemoglobin content (CHr) significantly increased on Day 4, but decreased with a significant increase in reticulocyte count from Day 7. Log_10_(serum ferritin) significantly decreased after Day 11. Log_10_(EPO concentration) was lower at all time points. Compared with the DA group, the Rox group showed significant differences in all variables except CHr. These results suggest that Rox improves hematopoiesis and iron metabolism early after administration independent of EPO concentration.

## 1. Introduction

Anemia associated with impaired kidney function is defined as renal anemia, and its main cause has been established to be impaired erythropoietin (EPO) production in the kidney. Renal anemia in dialysis patients requires massive and frequent blood transfusions, which are associated with risks of infections such as hepatitis [1], organ damage due to iron deposition [2], and skin pigmentation [3], resulting in significant impairment of the quality of life (QOL). Following the successful cloning of the EPO gene in 1985 [4], human recombinant erythropoietin (rHuEPO) was developed and has been available in Japan for dialysis patients since 1990 and for patients with chronic renal failure since 1994, providing outstanding outcomes. Moreover, rHuEPO, now collectively known as erythropoietin-stimulating agents (ESAs), includes recently developed long-acting products that have demonstrated significant efficacy in improving anemia, preventing complications, and improving patients’ QOL [5,6]. Meanwhile, several clinical problems have been reported, including elevated blood pressure [7], pro-anemic effects caused by concomitant drugs, especially angiotensin-converting enzyme inhibitors [8], anti-rHuEPO effects [9], production of anti-EPO antibodies [10,11], hemoglobin cycling caused by intermittent administration of injectable formulations [12], and ESA-refractory anemia [13].

As a solution to these issues, a new therapeutic strategy has recently been gaining traction. More specifically, multiple hypoxia-inducible factor (HIF) stabilizers are being developed as drugs for a new mechanism of action to stimulate endogenous EPO production. In November 2019, roxadustat (Rox), a HIF prolyl hydroxylase domain (PHD)-containing protein inhibitor, was approved in Japan for the treatment of anemia in hemodialysis (HD) patients. HIF was discovered in 1992 as a transcription factor that is hypoxia-inducible and responsible for the regulation of EPO gene expression [14]. HIF is induced into the degradation process by PHD under normal oxygen concentrations, but under hypoxic conditions or in the presence of PHD inhibitors, PHD is inactivated and HIF induces the expression of its target genes. Rox stabilize HIF by inhibiting PHD, allowing HIF to act on EPO-producing cells in the kidney and liver to induce endogenous EPO production and subsequent hematopoiesis. Notably, in addition to its EPO-stimulating effect, HIF also regulates the expression of hematopoiesis-related genes, especially the genes of molecules involved in iron metabolism and utilization [15,16,17].

To maximize the efficacy of ESAs and HIF stabilizers in enhancing hematopoiesis, a sufficient supply of iron, the source of hemoglobin, is essential. Hepcidin, a peptide produced by the liver, is the key element for iron metabolism in vivo. Hepcidin regulates iron supply to the bloodstream by binding to and degrading ferroportin, a protein responsible for iron supply from cells to the bloodstream [18]. Daily hematopoiesis requires approximately 6–8 times the amount of iron present in the blood, most of which is provided by recirculation from reticuloendothelial macrophages. Therefore, good iron metabolism is essential for effective hematopoiesis. However, conventional ESAs and HIF stabilizers are supposed to differ on hematopoiesis and iron metabolism. The information on the differences between the two seems to be useful for anemia treatment.

In this study, we frequently measured the reticulocyte indices, which reflect the most recent status of hematopoiesis, and iron-related factors in patients who were switched from conventional ESAs to a HIF stabilizer, to investigate the effects of the HIF stabilizers on hematopoiesis and iron metabolism. We also compared the changes in the reticulocyte indices and iron-related factors over one and four weeks of treatment with an ESA, respectively, with those under treatment with the HIF stabilizers to examine the difference in the effects of the two agents on hematopoiesis and iron metabolism.

## 2. Results

### 2.1. Patients

Two patients in the Rox group were excluded during the observation period, one due to stroke and the other due to gastrointestinal hemorrhage. Thus, the remaining 30 Rox-treated patients were included in the analysis.

Mean age was 71.3 ± 12.4 years in the Rox group and 73.2 ± 10.5 years in the DA group. There were 17 men and 13 women in the Rox group and 8 men and 7 women in the DA groups. Median duration of HD therapy was 7.2 [2.8–16.6] and 18.3 [2.42–26.8] years, respectively. The most common primary diagnoses were chronic glomerulonephritis (*n* = 7 and 6 in the Rox and DA groups, respectively), diabetic nephropathy (*n* = 14 and 3), and renal sclerosis (*n* = 8 and 5). Mean red blood cell (RBC) counts (×104/μL) in the Rox and DA groups were 345.0 ± 26.9 and 345.0 ± 42.1; mean hemoglobin (Hb) levels (g/dL) were 10.6 ± 0.7 and 10.4 ± 0.7; mean reticulocyte (Ret) counts (×104/μL) were 41.4 ± 15 and 35.3 ± 12.8; and mean CHr levels (pg) were 32.4 ± 1.7 and 32.2 ± 1.9, respectively. Median serum ferritin levels (s-ft, ng/mL) in each group were 48.2 (31.4–97.8) and 57.6 (34.3–96.8); mean serum iron levels (s-Fe, μg/dL) were 63.5 ± 22.4 and 54.9 ± 11.2; mean total iron-binding capacity (TIBC, μg/dL) were 254.2 ± 32.3 and 243.2 ± 34.2; mean transferrin saturation (TSAT) (%) were 25.2 ± 8.4 and 22.8 ± 4.8; and median hepcidin levels (ng/mL) were 26.8 (12.8–54.4) and 15.4 (5.2–40.2), respectively. Median EPO concentrations (mIU/mL) in each group were 11.6 (6.6–17.6) and 9.2 (7.0–13.8), respectively. Mean dose of DA in the Rox group before the switch was 14.7 μg/week, and that in the DA group was 17.0 ± 13.8 μg/week. In the Rox group, 19 and 11 patients received 70 and 100 mg of Rox per administration, respectively. There was no significant difference in any of the demographic variables between the DA and Rox groups (Table 1).

### 2.2. Change over Time in EPO Concentration in the Rox and the DA Group

In the Rox group, the mean Log_10_(EPO concentration) was lower than the Day-0 value at all time points, with significant differences on Day 14 (*p* < 0.01 vs. Day 0) and Day 28 (*p* = 0.047 vs. Day 0). On the other hand, log_10_(EPO concentration) increased sharply immediately after DA administration and then gradually declined, with the Day-7 value being similar to the Day-0 value (*p* < 0.01 at Day 0 post-HD and Days 2 and 4 vs. Day 0; Figure 1).

### 2.3. Changes over Time in Erythrocyte Indices in the Rox Group

Hb levels showed a significant increase on Day 28 (*p* < 0.01 vs. Day 0). RBC count showed a significant increase on Day 21 (*p* = 0.03 vs. Day 0) and Day 28 (*p* < 0.01 vs. Day 0). No significant changes were observed in mean corpuscular volume or mean corpuscular Hb.

Ret count showed a slight increase on Day 2 (*p* = 0.01, vs. Day 0) and then dropped to baseline on Day 4. The count showed significant increases on Day 7 and all time points thereafter, with the highest value on Day 9 (*p* < 0.01 on Days 7, 9, 11, 14, 21, 28 vs. Day 0; Figure 2).

### 2.4. Correlation between Reticulocytes and Red Blood Cell Counts

We investigated the relationship between Ret count and RBC count to examine whether the increase of Ret count affects RBC count. Ret represents real-time hematopoiesis, on the other hand the increase of RBC count takes a time since reflecting hematopoiesis because it has a lifespan of 120 days. So, for analysis, Ret count used the value of Day 9, the highest value during the course of this study, and RBC used the value of Day 28. Ret count on Day 9 and RBC count on Day 28 exhibited a significant positive correlation (*r* = 0.005 (95%CI: 0.000 to 0.009; *p* = 0.04)). Moreover, the correlation between increase of Ret count from day 0 to day 9 and that of RBC count from day 0 to day 28 exhibited a significant positive relation (*r* = 0.005 (95%CI: 0.002 to 0.008; *p* = 0.001)) (Figure 3).

### 2.5. Changes over Time in CHr, Hepcidin Levels and Iron-Related Indices in the Rox Group

CHr showed a significant increase on Day 4 (*p* < 0.01 vs. Day 0), but then declined with no significant difference at any time point compared with Day 0. Log_10_(hepcidin) showed a decreasing trend over time, with significant differences on Day 2 and all time points thereafter (*p* < 0.01 on Days 2, 4, 7, 11, 14, 21, and 28 vs. Day 0).

The s-Fe levels showed a significant increase on Day 2 (*p* = 0.02 vs. Day 0) and then declined, with a significant decrease on Day 7 (*p* < 0.01 vs. Day 0). The levels again increased on Day 9 and then again showed a decreasing trend. TSAT showed a similar pattern of change to s-Fe but showed no significant difference on Day 2 and showed significant decreases on Days 7, 14, 21, and 28 (*p* < 0.01 vs. Day 0). TIBC showed significant decreases on Day 2 and all time points thereafter (*p* < 0.01 on Days 2, 4, 7, 11, 14, 21, and 28 vs. Day 0), and Log_10_(s-ft) showed significant decreases after Day 11 (*p* = 0.02 on Day 11 vs. Day 0; *p* < 0.01 on Days 14, 21, and 28 vs. Day 0; Figure 4).

### 2.6. Comparison between the Rox 70 mg and 100 mg Groups

Changes over time in various variables were compared between the Rox 70 mg and Rox 100 mg groups. There was no significant difference in response of erythrocyte indices and EPO concentration between the groups (Hb; *p* = 0.66, RBC count; *p* = 0.78, MCV; *p* = 0.92, MCH; *p* = 0.37, Log_10_(EPO concentration); *p* = 0.27, Ret count; *p* = 0.57, Appendix A). Moreover, CHr, hepcidin and iron-related indices showed no significant difference between the groups (CHr; *p* = 0.40, Log_10_(hepcidin); *p* = 0.50, s-Fe; *p* = 0.26, TSAT; *p* = 0.26, TIBC; *p* = 0.07, Log_10_(s-ft); *p* = 0.39, Appendix A).

### 2.7. Correlation between s-ft and Hepcidin Levels

On Day 0, log_10_(s-ft) and log_10_(hepcidin) showed a significant positive correlation (*r* = 0.61 (95% CI: 0.32 to 0.80; *p* < 0.01)), but the correlation was weaker on Day 28 (*r* = 0.41 (95% CI: 0.06 to 0.67; *p* = 0.02); Figure 5).

### 2.8. Comparison between the Rox and DA Groups

Changes over time in various variables were compared between the Rox and DA groups; Ret, CHr, and log_10_(hepcidin) were compared at every HD visit up to Day 7, and iron-related factors were compared at the above time points plus Days 14 and 28.

There was a significant difference in Ret count on Day 7 between the two groups (*p* = 0.001), with the Rox group showing an increase, whereas the DA group did not. There was no significant difference in CHr between the two groups (*p* = 0.99). Log_10_(hepcidin) showed a decreasing trend up to Day 7 in the Rox group, whereas it decreased up to Day 4 and returned to baseline on Day 7 in the DA group (*p* = 0.002).

Both s-Fe and TSAT increased on Day 2 and then declined in both groups, showing a greater decrease in the Rox compared with the DA group on Day 7 and thereafter, resulting in a significant difference between the two groups (*p* = 0.001 for s-Fe, *p* < 0.001 for TSAT). TIBC showed no change in the DA group, whereas it started to increase from Day 2 in the Rox group (*p* < 0.001). Log_10_(s-ft) remained unchanged in the DA group, whereas it started to decrease from Day 7 in the Rox group (*p* < 0.001; Figure 6).

## 3. Discussion

This study involved 30 patients on outpatient HD therapy who were initially treated with DA and then switched to Rox. We measured mature erythrocytes, reticulocyte indices, and iron-related factors at every HD visit during the first two weeks after the treatment switch (Days 0–14) and again on Days 21 and 28. We measured EPO concentration every week and examined their changes from Day-0 values. DA is a once-weekly ESA and has been shown to improve iron utilization to the maximum 3–4 days post-dose and then to return to near baseline by one week post-dose [19], suggesting that a one-week hematopoiesis cycle is repeated with each DA dose. By comparison, oral treatment with Rox three times per week is expected to induce endogenous EPO production, leading to continuous hematopoiesis [20].

In this study, EPO concentration in the DA group sharply increased after administration and then declined down to about one-seventh on Day 2, and to baseline on Day 7. Previous studies have shown that the mean EPO concentration after Rox administration reaches a maximum of 130 mIU/mL at about 12 h, decreases to one-third at 24 h, and returns to almost baseline at 48 h [20,21]. It can therefore be speculated that the blood EPO concentrations in the DA and Rox groups would show completely different values and patterns of change. Furthermore, HIF has been shown to improve iron metabolism independent of its effects on EPO [15], suggesting that Rox and DA have different effects on hematopoiesis and iron metabolism. However, previous studies have measured these parameters only every 1–2 weeks, and there has been no study examining the effects of Rox on hematopoiesis and iron metabolism in detail.

In the present study, significant increases in Hb and RBC were observed on Day 28 and Days 21 and 28, respectively, demonstrating that switching treatment to Rox resulted in improvement of anemia. In addition, the Ret count, a reticulocyte index reflecting the most recent hematopoietic capacity, showed a sustained increase on and after Day 7, with the highest value on Day 9. Erythroid progenitor cells stimulated by EPO differentiate into pre-erythroblasts, erythroblasts, reticulocytes, and mature erythrocytes. Differentiation from pre-erythroblasts to reticulocytes takes 4–6 days, and reticulocytes remain in the bone marrow for 2–3 days before being released into the peripheral blood to become mature erythrocytes in 1–2 days. Therefore, considering the number of days required for the differentiation process, Rox seems to have exerted a strong hematopoietic effect immediately after administration, leading to an improvement in anemia. The reason it took longer before this effect was reflected in Hb and RBC levels is likely because mature erythrocytes have a long lifespan of 120 days. In this study, Ret count and the amount of increase on Day 9 are correlated with RBC count and the amount of increase on Day 28.

For the evaluation of iron metabolism and utilization, we focused on CHr and hepcidin levels. Hemoglobin synthesis takes place mainly in erythroblasts. Iron, which is bound to and transported by transferrin in the blood, is taken up into erythroblasts via binding to transferrin receptors expressed on the plasma membrane of the cells and is used as a major constituent of hemoglobin. CHr is measured by applying the principle of flow cytometry and indicate the mean hemoglobin content of neonatal reticulocytes. Thus, it considered to directly reflect iron sufficiency at the hematopoietic level. Hepcidin is a major regulator of in-vivo iron metabolism, a liver-derived peptide that is induced by iron and pro-inflammatory signals [22,23], and is suppressed by EPO and HIF [15,24].

In the present study, we observed a significant decrease in hepcidin and a significant increase in serum iron levels on Day 2 after Rox administration and a significant increase in CHr on Day 4. Previous studies examining the change in CHr after iron supplementation also showed that the highest level was reached on Day 4 [25,26]. The observation that hepcidin levels significantly decreased from Day 2 together with the number of days required for the differentiation into reticulocytes suggest that Rox suppresses hepcidin and increases serum iron levels immediately after administration, thereby improving iron utilization for hemoglobin synthesis. We also observed a significant increase in TIBC from Day 2. An in vivo study has demonstrated that HIF induces transferrin [27]. TIBC, which is considered the second most reliable indicator of iron stock after s-ft [28], significantly increased from a time when there was no change in s-ft, suggesting the effect of HIF.

The initial daily dose was 70 mg or 100 mg depending on the amount of DA administered. However, there was no significant difference between the two groups in response of all variables measured in this study. It was suggested that Rox switching method according to the package insert is expected to have similar reactions in hematopoiesis and iron metabolism between the two groups.

Moreover, the significant positive correlation between hepcidin and s-ft seen on Day 0 was also attenuated on Day 28. Hepcidin has also been shown to be significantly correlated with s-ft under ESA use in previous studies [29,30]. Some reports suggest that the inhibitory effect of HIF on hepcidin is mediated by EPO, whereas others suggest that it is a direct action [16,17]. In the present study, a strong suppression of hepcidin was observed despite low EPO concentrations, and there was a weakened effect of s-ft, suggesting the possibility of an EPO-independent effect of HIF.

In the comparison between the Rox and DA groups, variables for the real-time monitoring of hematopoiesis and iron metabolism were measured up to Day 7, considering the hematopoiesis cycle in the DA group. Iron-related factors were also measured on Days 14 and 28 to determine whether there was any difference in iron utilization. Comparisons of the reticulocyte indices, hepcidin, and iron-related factors showed significant differences in all variables except CHr. In the DA group, there was no change in either Ret count or TIBC, whereas in the Rox group, significant increases were observed in Ret count from Day 7 and in TIBC from Day 2. Hepcidin returned to baseline on Day 7 in the DA group, whereas it showed a sustained decline on Day 7 in the Rox group. Among iron-related indices, s-ft and TSAT showed a greater decrease in the Rox group compared with the DA group in the second half of the observation period, suggesting an increase in iron consumption. Apart from its inhibitory effect on hepcidin, HIF has also been shown to induce heme-oxygenase-1 [31], ceruloplasmin [32], ferroportin [33], and transferrin receptor [34], which are essential for iron recirculation. Therefore, the increased iron consumption observed in the Rox group may be attributable not only to enhanced hematopoiesis, but also to improved iron utilization efficiency, mainly due to the suppression of hepcidin activity.

However, this study was conducted at a single center, with a small sample size and a limited observation period early after the treatment switch. Another possible limitation is that patients’ iron levels are maintained at low levels at our hospital, which may have affected the iron metabolism-related data. Further studies are needed to accumulate more cases, analyze data on longer-term use, and determine the optimal dose of iron supplementation in patients treated with HIF-related drugs.

## 4. Materials and Methods

### 4.1. Patients

This study involved patients receiving outpatient HD therapy at our hospital who were treated with DA for renal anemia and consented to participate in the study, including 32 patients who switched to Rox (Rox group) and 15 patients who continued to receive DA (DA group). Patients with fundal hemorrhage in the Rox group and those with malignancy in both groups were excluded. All patients were followed for 4 weeks and received 3 sessions of HD therapy per week, each lasting for 4–5 h.

Anemia treatment was provided with a target Hb level of 10–12 g/dL according to the guidelines of the Japanese Society of Dialysis Therapy. The dose of DA was adjusted accordingly. Rox administration was continued at the initial dose according to the package insert, and no iron supplement or iron-containing phosphate adsorbent was used during the observation period.

All subjects gave their informed consent for inclusion before they participated in the study. The study was conducted in accordance with the Declaration of Helsinki, and the protocol was approved by the Ethics Committee of Biomarker Society, Inc., which consists of 7 committees, including outside experts (approval number 2020-01, 25 March 2020).

### 4.2. Rox Group

Rox administration was started 1 week after the last dose of DA. The daily dose of Rox was 70 mg in patients who had received DA at doses < 20 μg/week (Rox 70 mg group) and 100 mg in those who had received DA at doses ≥ 20 μg/week (Rox 100 mg group), following the directions on the package insert, and was administered at each visit for three-times-weekly HD therapy.

Blood samples were collected at the start of HD, at every HD visit between the first day of treatment with Rox (Day 0) and Day 14, and then on Days 21 and 28 (9 times in total) to measure mature erythrocytes, reticulocyte indices (Ret count, CHr), iron-related factors, and hepcidin. TSAT was calculated as s-Fe concentration divided by total iron-binding capacity multiplied by 100. EPO concentration was also measured on Days 0, 7, 14, 21, and 28.

### 4.3. DA Group

DA was administered at the end of the first HD session of the week. With the day of DA administration defined as Day 0, mature erythrocytes, reticulocyte indices, iron-related factors, hepcidin, and EPO concentration were measured in the same manner as in the Rox group between Days 0 and 7. Iron-related factors were also measured on Days 14 and 28; EPO concentration was also measured immediately after each DA administration (Day 0 post-HD).

Erythrocyte lineage cells were counted using an ADVIA2120 hematology analyzer (Siemens Healthcare Diagnostics, Tarrytown, NY), EPO concentration was measured by chemiluminescent enzyme immunoassay, and hepcidin levels were measured by the quantitative method of liquid chromatography coupled with tandem mass spectrometry [21].

### 4.4. Statistical Analysis

Analyses were performed with the SAS system ver. 9.3 software (SAS Institute, Cary, NC). Data are presented as the mean ± SD and the median with interquartile range. The *t*-test was used to compare groups in terms of normally distributed continuous variables, and the Mann-Whitney U test was used for other skewed continuous variables. The chi-square test was used to compare nominally scaled variables. One-way repeated measures analysis of variance was performed for changes over time, and Bonferrroni’s multiple comparison was performed for post-hoc tests. To evaluate the relationship between Ret and RBC count, hepcidin, and s-ft, the Pearson product-moment correlation coefficient and univariable linear regression model were used. The values of hepcidin, s-ft and EPO concentration, which were not normally distributed, were log-transformed before performing the above parametric analysis. Two-way repeated measures analysis of variance was used for comparison between Rox 70 mg group and Rox 100 mg group, the Rox and DA groups. Two-tailed P values less than 0.05 were considered to indicate a statistically significant difference.

## 5. Conclusions

The results of this study suggest that administration of Rox enhances hematopoiesis and improves iron metabolism early after switching treatments, resulting in increased iron consumption compared with patients who continued DA. Our data also suggest that these effects of Rox are independent of EPO concentration.

Shortened erythrocyte lifespan and functional iron deficiency due to chronic inflammatory conditions have been identified as problems in treating anemia in HD patients. Information on the differences in the effects of Rox and conventional ESAs on hematopoiesis and iron metabolism may help select a suitable anemia treatment for individual HD patients.

## Figures and Tables

**Figure 1 ijms-21-07153-f001:**
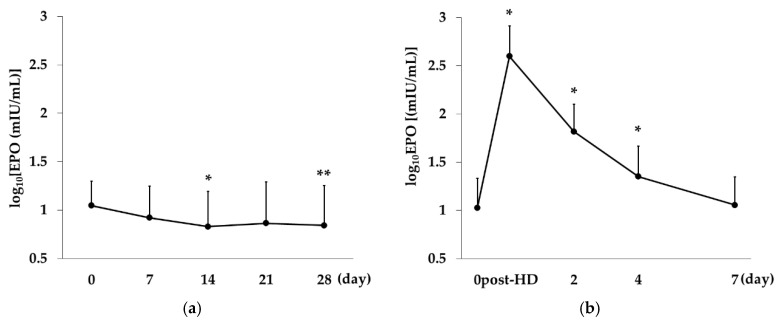
Change over time in erythropoietin concentration in the Rox and the DA group. (**a**) In the Rox group, log_10_(EPO concentration) showed significant decreases on Day 14 (*p* < 0.01 vs. Day 0) and Day 28 (*p* = 0.047 vs. Day 0). (**b**) In the DA group, EPO concentration increased sharply immediately after DA administration and then gradually declined, returning to baseline on Day 7. One-way repeated measures analysis of variance and Bonferrroni’s multiple comparison were performed for post-hoc tests. The data are represented as mean ±SD. * *p* < 0.01, ** *p* < 0.05.

**Figure 2 ijms-21-07153-f002:**
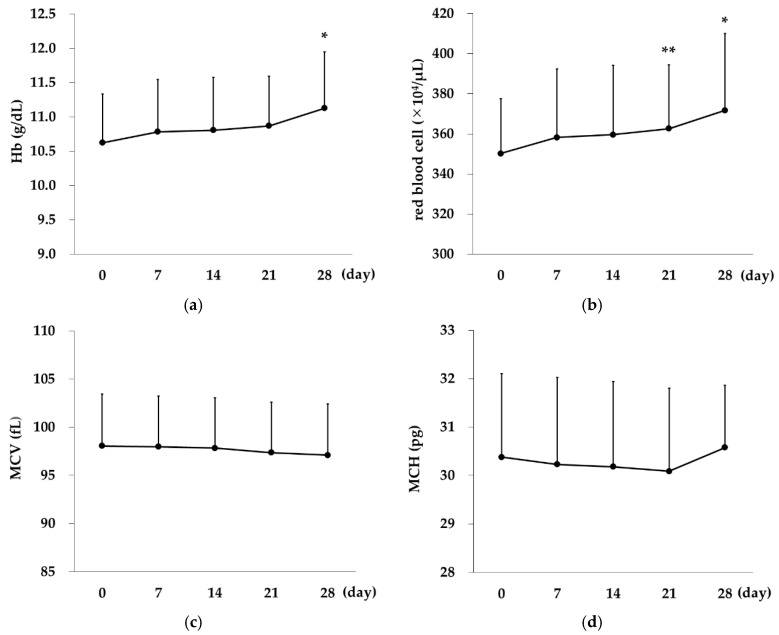
Changes over time in erythrocyte indices and EPO concentration in the Rox group. (**a**) Hb levels showed a significant increase on Day 28 (*p* < 0.01 vs. Day 0). (**b**) RBC count showed significant increase on Day 21 (*p* = 0.03 vs. Day 0) and Day 28 (*p* < 0.01 vs. Day 0). (**c**) Mean corpuscular volume (MCV), (**d**) Mean corpuscular Hb (MCH), and (**e**) Ret count showed significant increases on Day 7 and all time points thereafter (*p* < 0.01 on Days 7, 9, 11, 14, 21, 28 vs. Day 0). One-way repeated measures analysis of variance and Bonferrroni’s multiple comparison were performed for post-hoc tests. The data are represented as mean ±SD. * *p* < 0.01; ** *p* < 0.05.

**Figure 3 ijms-21-07153-f003:**
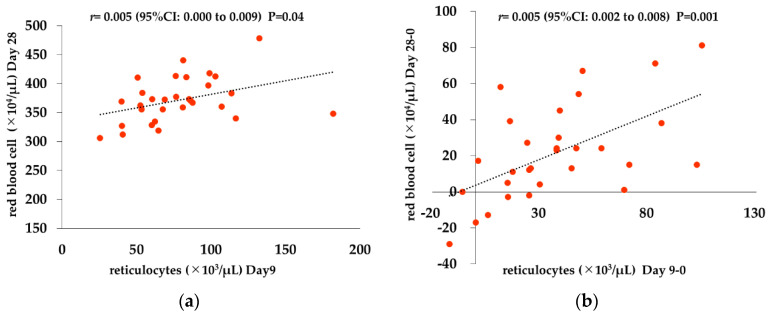
Correlation between reticulocytes and red blood cell counts. (**a**) Ret count on Day 9 and RBC count on Day28: Ret and RBC counts showed a significant positive correlation (*r* = 0.005 [95%CI: 0.000 to 0.009; *p* = 0.04]). (**b**) Increase of Ret count from day 0 to day 9 and that of RBC count from day 0 to day 28: Increase of Ret and RBC counts showed a significant positive correlation (*r* = 0.005 [95%CI: 0.002 to 0.008; *p* = 0.001]). Pearson product-moment correlation coefficient and univariable linear regression model were used.

**Figure 4 ijms-21-07153-f004:**
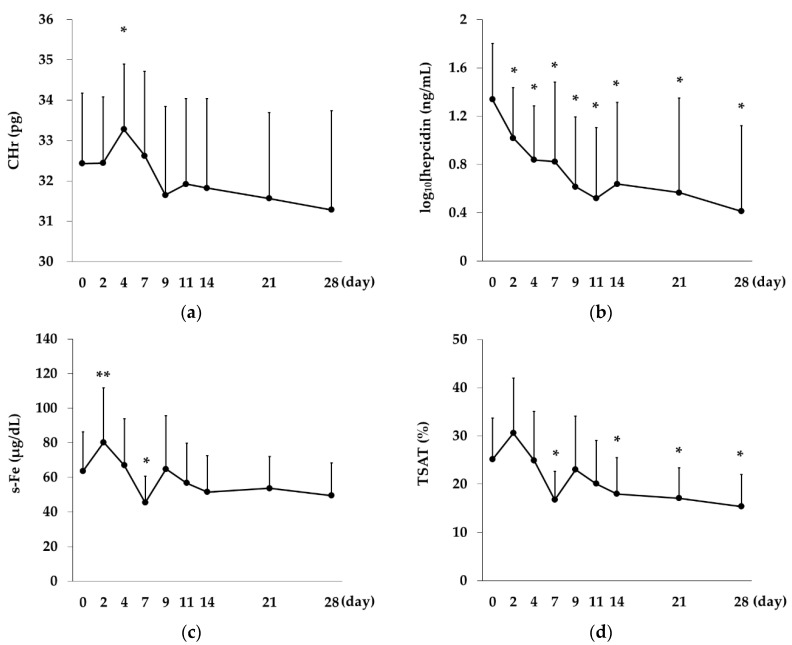
Changes over time in CHr, hepcidin levels and iron-related indices in the Rox group. (**a**) CHr showed a significant increase on Day 4 (*p* < 0.01 vs. Day 0). (**b**) Log_10_(hepcidin) showed significant decreases on Day 2 and all time points thereafter (*p* < 0.01 on Days 2, 4, 7, 11, 14, 21, and 28 vs. Day 0). (**c**) s-Fe showed a significant increase on Day 2 (*p* = 0.02 vs. Day 0) and a significant decrease on Day 7 (*p* < 0.01 vs. Day 0). (**d**) TSAT showed significant decreases on Days 7, 14, 21, and 28 (*p* < 0.01 vs. Day 0). (**e**) TIBC showed significant increases on Day 2 and all time points thereafter (*p* < 0.01 on Days 2, 4, 7, 11, 14, 21, and 28 vs. Day 0). (**f**) Log_10_(s-ft) showed significant decreases after Day 11 (*p* = 0.02 on Day 11; *p* < 0.01 on Days 14, 21. and 28 vs. Day 0). One-way repeated measures analysis of variance and Bonferrroni’s multiple comparison were performed for post-hoc tests. The data are represented as mean ±SD. *, *p* < 0.01; **, *p* < 0.05.

**Figure 5 ijms-21-07153-f005:**
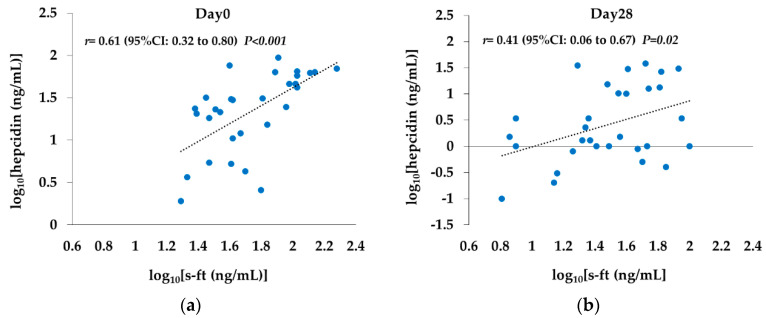
Correlation between s-ft and hepcidin levels. (**a**) Day 0: Log_10_(s-ft) and Log_10_(hepcidin) showed a significant positive correlation (*r* = 0.61 [95% CI: 0.32 to 0.80; *p* < 0.01]). (**b**) Day 28: The correlation was weaker compared with Day 0 (*r* = 0.41 [95% CI: 0.06 to 0.67; *p* = 0.02]). The Pearson product-moment correlation coefficient and univariable linear regression model were used.

**Figure 6 ijms-21-07153-f006:**
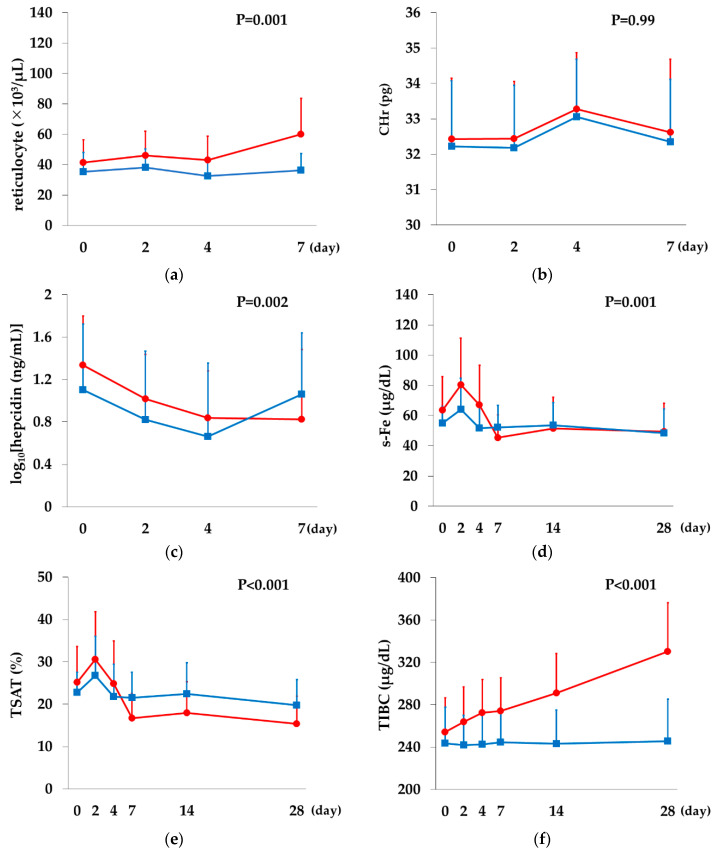
Comparison between the Rox and DA groups. (**a**) Ret count showed a significant difference between the groups (*p* = 0.001). (**b**) CHr showed no significant difference between the groups (*p* = 0.99). (**c**) Log_10_(hepcidin) showed a significant difference between the groups (*p* = 0.002). (**d**) s-Fe showed a significant difference between the groups (*p* = 0.001). (**e**) TSAT showed a significant difference between the groups (*p* < 0.001). (**f**) TIBC showed a significant difference between the groups (*p* < 0.001). (**g**) Log_10_(s-ft) showed a significant difference between the groups (*p* < 0.001). Two-way repeated measures analysis of variance was performed. The data are represented as mean ± SD. (
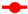
 Rox, 
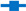
 DA).

**Table 1 ijms-21-07153-t001:** Characteristic.

Variables	Rox			DA	*p* Value *
	All	(70 mg)	(100 mg)		
N	30	19	11	15	
Age (years)	71.3 ± 12.5	69.4 ± 13.1	77.0 ± 10.2	73.2 ± 10.5	0.95
Gender					1
Men	17	13	4	8	
Women	13	6	7	7	
Duration of dialysis (years) †	7.2 (2.8–16.6)	9.2 (2.8–18.5)	3.3 (3.0–10.5)	18.3 (2.4–26.8)	0.29
Primary diagnosis					0.44
Chronic glomerulonephritis	7	5	2	6	
Diabetes nephropathy	14	10	4	3	
Renal sclerosis	8	4	4	5	
Other	1	-	1	1	
Kt/V	1.50 ± 0.20	1.58 ± 0.17	1.46 ± 0.14	1.50 ± 0.23	0.93
Red blood cells (×10^4^/μL)	350.2 ± 26.9	354.6 ± 27.5	342.5 ± 22.1	345.0 ± 42.1	0.63
Hemoglobin (g/dL)	10.6 ± 0.7	10.8 ± 0.6	10.3 ± 0.8	10.4 ± 0.7	0.37
MCV (fL)	98.1 ± 5.4	98.1 ± 4.4	98.0 ± 6.0	97.2 ± 5.8	0.62
MCH (pg)	30.4 ± 1.7	30.6 ± 1.6	30.1 ± 1.8	30.4 ± 2.1	0.92
Reticulocytes (×10^3^/μL)	41.4 ± 15.1	35.0 ± 8.7	52.4 ± 17.6	35.3 ± 12.8	0.34
CHr (pg)	32.4 ± 1.7	32.8 ± 1.7	31.8 ± 1.5	32.2 ± 1.9	0.72
serum-Ferritin (ng/mL) †	48.2 (31.4–97.8)	68.4 (40.2–99.9)	41.0 (26.3–64.2)	57.6 (34.3–96.8)	0.97
Iron (μg/dL)	63.5 ± 22.4	67.8 ± 23.3	56.1 ± 16.0	54.9 ± 11.2	0.20
TIBC (μg/dL)	254.2 ± 32.3	246.5 ± 32.3	267.5 ± 27.5	243.5 ± 34.2	0.32
Transferrin saturation (%)	25.2 ± 8.4	27.3 ± 7.9	21.4 ± 7.8	22.8 ± 4.8	0.38
Hepcidin (ng/mL) †	26.8 (12.8–54.4)	29.9 (18.2–62.6)	20.2 (4.7–31.2)	15.4 (5.2–40.2)	0.28
Albumin (g/dL)	3.5 ± 0.3	3.5 ± 0.3	3.6 ± 0.4	3.4 ± 0.3	0.46
C-reactiveprotein (mg/dL) †	0.12 (0.07–0.60)	0.10 (0.08–0.33)	0.19 (0.07–0.97)	0.19 (0.08–0.33)	0.91
erythropoietin (mIU/mL) †	11.6 (6.9–17.6)	9.7 (6.7–11.6)	18.5 (12.9–28.7)	9.2 (7.0–13.8)	0.39
Darbepoetin α (IU/week)	18.3 ± 14.7	9.5 ± 3.0	33.6 ± 15.0	17.0 ± 13.8	0.83

†, Mean ± SD and median and interquartile range (IQR); *p* value *: Rox (All) vs. DA. Abbreviations: MCV, mean corpuscular volume; MCH, mean corpuscular hemoglobin; CHr, reticulocyte hemoglobin content; TIBC, total iron-binding capacity.

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
