# Peer review of "A Hypoxia-Inducible Factor Stabilizer Improves Hematopoiesis and Iron Metabolism Early after Administration to Treat Anemia in Hemodialysis Patients"

_ijms, 2020, doi:10.3390/ijms21197153_

Round 1

Reviewer 1 Report

This manuscript describes the differences between two (or three) treatments of hemodialysis patients. The comparison is between DA (darbepoetin), which was the treatment they were already receiving, and Rox (Roxadustat), a HIF stabilizer. They observed a reduction in EPO concentration, hepcidin concentration and a clear increase in total iron binding capacity. They conclude that Rox improves hematopoiesis and iron metabolism.

Although interesting, a few improvements should be made for publication to be considered:

Referencing in the introduction section is incomplete. Many statements without appropriate reference.

The 30 Rox patients clearly are two groups, those that received 70mg and those that received 100mg, based on previous treatment. The Rox patients should therefore be subdivided in Table 1 and potentially in the other figures as well. Alternatively, it should be shown that there are no statistical differences between the 70mg and 100mg groups in their response.

It should be clearly stated which statistical test was use for which results. Also, it is not mentioned what the error bars represent.

A direct comparison of reticulocyte and red blood counts at the later time points should be included.

Reviewer 2 Report

Review of manuscript ijms-931804

The manuscript by C. Ogawa ad colleagues describes the results of a study aiming at comparing the efficacy of Roxadustat, an HIF stabilizer, on hemodialysis-induced anemia and iron deficiency, with a group of patients treated by darbepoetin, a classical EPO stimulating agent. The efficacy of each treatment is evaluated through classical parameters of erythropoiesis (EPO level in the plasma, erythrocytes in the blood stream, hemoglobin, MCV, HCT, …) and of iron metabolism (serum ferritin, serul iron, TIBC, transferrin saturation, hepcidin, …). Results indicate that some of the erythropoiesis parameters are ameliorated, as well as some of iron metabolism parameters. Overall, the authors conclude that Rox treatment improve the anemia status of hemodialysis patients.

The manuscript is interesting, well written, results are sound and sustain authors’ conclusions. However, some points need authors’ attention:

  1. This is a mono-center study, with a low number of patients. As such, the limitations of the study must appears earlier in the discussion section. This low number of patients may explain the relatively large standard deviations in the results. However, the statistical treatment of the data is good, so that the interpretation of the results is clear and sound.
  2. However, within the ROX treatment group, there are two groups of treatment, one with 70 mg per week and one with 100mg per week. There is no mention in the manuscript of a possible difference in the results between these two groups. Such a difference may influence the results of the comparison with DA treatment.
  3. A direct comparison of EPO concentration between the two groups should be of interest for the reader. It is suggested to add the EPO results from ROX patients in figure 1, and to provide a statistical comparison between the two groups of patients.
  4. As well, some of the results in figure 5 are also presented in figure 2 and 3. This should be optimized. Moreover, both the X-axis scale and the Y-axis scale are different between figure 5 and other figures, which renders the comparison difficult for the reader. As a whole, the presentation of the results should be made by direct comparison of the two groups of patients, with a two-way ANOVA analysis, with allows detecting the influence of both time and treatment.
  5. Lines 192-194: please provide a reference for this statement.

Round 2

Reviewer 1 Report

Thank you for your alterations. I think the changes in the results section improve the clarity of the data.